# Cutaneous Melanoma and Hormones: Focus on Sex Differences and the Testis

**DOI:** 10.3390/ijms24010599

**Published:** 2022-12-29

**Authors:** Ilaria Cosci, Giuseppe Grande, Andrea Di Nisio, Maria Santa Rocca, Paolo Del Fiore, Clara Benna, Simone Mocellin, Alberto Ferlin

**Affiliations:** 1Veneto Institute of Oncology IOV-IRCCS, 35128 Padova, Italy; 2Unit of Andrology and Reproductive Medicine, University Hospital of Padova, 35128 Padova, Italy; 3Department of Medicine, University of Padova, 35128 Padova, Italy; 4Soft-Tissue, Peritoneum and Melanoma Surgical Oncology Unit, Veneto Institute of Oncology IOV-IRCCS, 35128 Padua, Italy; 5Department of Surgical, Oncological and Gastroenterological Sciences (DISCOG), University of Padua, 35128 Padova, Italy

**Keywords:** melanoma, immunotherapy, male fertility

## Abstract

Cutaneous melanoma, the most aggressive type of skin cancer, remains one the most represented forms of cancer in the United States and European countries, representing, in Australia, the primary cause of cancer-related deaths. Recently, many studies have shown that sex disparities previously observed in most cancers are particularly accentuated in melanoma, where male sex is consistently associated with an increased risk of disease progression and a higher mortality rate. The causes of these sex differences rely on biological mechanisms related to sex hormones, immune homeostasis and oxidative processes. The development of newer therapies, such as immune checkpoint inhibitors (ICIs) (i.e., anti–PD-1 and anti–CTLA-4 monoclonal antibodies) has dramatically changed the treatment landscape of metastatic melanoma patients, though ICIs can interfere with the immune response and lead to inflammatory immune-related adverse events (irAEs). Recently, some studies have shown a potential adverse influence of this immunotherapy treatment also on male fertility and testicular function. However, while many anticancer drugs are known to cause defects in spermatogenesis, the effects of ICIs therapy remain largely unknown. Notwithstanding the scarce and conflicting information available on this topic, the American Society of Clinical Oncology guidelines recommend sperm cryopreservation in males undergoing ICIs. As investigations regarding the long-term outcomes of anticancer immunotherapy on the male reproductive system are still in their infancy, this review aims to support and spur future research in order to understand a potential gonadotoxic effect of ICIs on testicular function, spermatogenesis and male fertility.

## 1. Introduction

The melanoma is a rare malignant skin cancer that arises from melanocytes, cells mainly residing at the level of the derma-epidermal junction, inside intercellular spaces formed by skin’s basal layer keratinocytes. Melanocytes originate from neural crest and they are found at basal layer of epidermis, in mucous membranes, in dermis, in hair follicles, in envelopes of the central nervous system and in the eyes vascular layer. The exposure to ultraviolet (UV) radiation, namely UVA (315–400 nm) and UVB (280–315 nm), seems to represent one of the major risk factor for melanoma development. It is estimated that about 70% of cancerous skin damage onset is caused by strong and continuous exposure to UV radiation, in particular to UVA which is much more abundant than UVB in sunlight, accounting for 95% of solar UV radiation. Moreover, UVA is the primary source of light used in indoor tanning beds and it can reach doses 12-times higher than those of the sun [1]. However, there are many factors, including hair color, skin type, genetic background, environment and history of tanning determining the skin’s response to UV radiations. About 15% of melanomas occurs in patients with a positive family history and approximately 22% of these cases are caused by a germline mutation in a tumor predisposing gene. *CDKN2A* gene mutations are responsible for the majority of hereditary melanomas; indeed, over half of individuals with multiple primary melanomas carry mutations in this gene. However, other susceptibility genes have been identified recently, including *CDK4*, *TERT*, *ACD*, *TERF2IP*, *POT1*, *MITF*, *MC1R*, and *BAP1* and an increased melanoma risk is observed in mixed cancer syndromes caused by mutations in *PTEN*, *BRCA2*, *BRCA1*, *RB1*, and *TP53* genes [2,3]. The USA American Surveillance, Epidemiology and End Results program (SEER) highlighted that melanoma still remains one of the most represented cancers in United States and in European countries, with an increasing incidence worldwide compared to other solid tumors. Australia has the highest incidence of melanoma in the world and melanoma represents the first cause of cancer-related death in subjects aged between 15 and 44 years [4]. According to GLOBOCAN, new melanoma cases in 2020 were estimated 324,635, including 173,844 in males, of which 25,975 in males aged between 20 and 50 years. Skin melanoma accounts for 1.7% of global cancer diagnosis, according to the latest SEER data, and it is the fifth most common cancer diagnosis in the US representing 5.6% of all cancer diagnoses. In 2021, 106,110 new cases of skin melanoma were estimated with a rate of 22.8 per 100,000 (both men and women) of new cases per year and a death rate of 2.2%. These rates are age-adjusted and based on 2014–2018 cases and 2015–2019 deaths [5].

## 2. Sex Disparities in Melanoma

Many studies documented disparities in cancer survival between sexes, including melanoma. Gender differences in melanoma progression have been studied by various authors since the late 1960s [6]. Epidemiological studies have showed that melanoma mortality is relatively constant among males and greater than females. Indeed, men have a twofold higher probability to develop malignant melanoma with a higher death rate [7]. Recently, a study investigated the impact of melanoma on patients’ life expectancy, according to tumor stage and sex, both in the single individual and at a population level, using US SEER data on a group of 104.938 subjects with a melanoma diagnosed from 2000 to 2017 [7,8]. The authors concluded that considering sex variation, females have a higher life expectancy than men. These findings evidenced that male sex is consistently associated with increased risk of melanoma and higher mortality and that is a major factor in survival differences, as previously demonstrated by research within North Europe and American populations [9,10]. A study performed on data collected between 2004 and 2005 from the SEER database, including 115,576 patients of which 62,938 males and 52,638 females, highlighted that men had overall lower cancer survival rate compared to women at the same melanoma stage evaluated according to 8^th^ edition of the AJCC staging system. The survival rates of males and females in each stage from IA–IIA were significantly different [11]. Nevertheless, although many studies clearly indicate that women with melanoma have a better prognosis than men, the underlying mechanisms are poorly understood [12,13]. Previous findings indicate that these gender differences are more likely due to biological reasons rather than lifestyle, socio-economic and behavioral factors. Although women are more likely to be careful regarding health care and prevention than men, this reason is not sufficient to explain the strong sex dependence variations in the incidence and mortality by cutaneous melanoma. Accordingly, various possible biological causes have been studied, such as immune homeostasis and function, oxidative stress response and X-linked genes. However, an additional hypothesis suggests a potential role of hormones in these differences [8,14,15,16]. For the first time in 1978, Shaw et al. [17] reviewed a series of 1861 patients with malignant melanoma to determine if there were endocrine influences on the disease survival, and concluded that hormones might have a role in metastasis formation and in the distribution of anatomical sites of primary lesions.

In a review of 1984, Rampen et al. [18] analyzed the role of hormones in malignant melanoma, namely considering six factors: location of the primary melanoma, stage of the disease at presentation, endocrine factors, immunologic factors, pattern of metastatic spread (i.e., lymphogenic versus hematogenic), environmental, and behavioral characteristics. They confirmed that the incidence of metastatic disease at the time of diagnosis is higher in men. Moreover, men tend to have an equal or shorter history before treatment, yet they have more advanced disease at the time of diagnosis. Men have an unfavorable outcome irrespectively of lesion site, tumor thickness, histogenetic subtype, and clinical stage of disease. These data suggest that the disease develops more rapidly in men. Thus, the aggressiveness and metastatic potential of cutaneous melanoma is more distinct in the male sex. These authors concluded that malignant melanoma might be a hormone-responsive tumor. Although the exact nature of such endocrine factors is still uncertain, the authors suggested the need to study hormone receptor mechanisms for the elucidation of the role of endocrine factors in melanoma behavior. The next paragraphs examine the possible contribution of sex hormones (estrogens and testosterone) in the biology and clinic of melanoma.

The relation between menopausal status and melanoma survival is conflicted. Reproductive status in women is characterized by fluctuations of sex steroid hormones. During the reproductive-age, women have a more reactive inflammatory profile and higher levels of T lymphocytes when compared to post-menopausal women [19,20]. However there appears to be no association between pregnancy and melanoma and the correlation between pre- and post-menopause and survival is contradictory. Indeed some studies and, recently, Enninga E.A.L., et al., have found no evidence of differences in post-menopausal groups, while others found significant differences in post-menopausal groups documented [21].

### 2.1. Melanoma and Estrogens

Sex hormones belong to the steroid hormone family, mainly synthesized by the adrenal cortex and gonads, and in minor part by other peripheral tissues, such as the skin. Indeed, skin cells contain the entire biochemical machinery for production of estrogens, vitamin D, testosterone (T) and dihydrotestosterone (DHT) either from precursors or, alternatively, through the conversion of cholesterol to pregnenolone and its subsequent transformation to biologically active steroids [22]. Estrogens regulate the growth and differentiation of normal and several neoplastic tissues (such as breast, ovarian and endometrial tumors). In particular, estrogens exert their effect through specific nuclear receptors α (ER-α), β (ER-β) and G protein-coupled estrogen receptor (GPER) on the cell membrane. Indeed, cutaneous ER levels are generally higher in women than in men. Despite ER-α and ER-β have identical general structure they have major differences in ligand-binding domain which is responsible of different final effects. For example, while ER-α, in breast tissue, after binding to estrogen, stimulates cell proliferation, ER-β acts with an inhibitory effect impairing directly G (2)/M checkpoint signaling in the cell cycle inducing an increasing apoptotic activity [23]. Furthermore, as confirmed by a study of Roger et al. ER-β has an antagonizing role on the proliferative action of ER-α [24]. The demonstrated that the expression of ER-β markedly decreases in the early stages of mammary carcinogenesis, confirming a protective effect of ER-β against the mitogenic effect of estrogens in human premalignant lesions. These results were in line with that observed in mice in which the ER-β gene has been inactivated in order to define the functions of ER-β receptor both in the normal and malignant breast tissue. In fact, studies in ER-β knockout mice (βERKO) revealed an abnormal epithelial growth, the overexpression of Ki67 and severe cystic breast disease, indicating a stimulatory role of ER-α and an inhibitory effect of ER-β in the proliferation of different estrogen-responsive tissues [25]. Subsequently, other clinical studies showed that ERβ have a suppressive effect on tumor progression in patients with breast, prostate, colon, and ovary cancers and a therapeutic potential in management of these malignant [26]. De Giorgi et al., evaluated ER expression in human melanoma tissues and in the adjacent healthy tissue to investigate whether the ERα:ERβ ratio had a role in neoplastic progression. Using quantitative reverse transcriptase–polymerase chain reaction and immunohistochemical analysis, they analyzed ERα and ERβ messenger RNA (mRNA) and ER protein expression from 14 patients, 12 with cutaneous melanoma (six women and six men) and two with melanocytic nevi (one woman and one man). This study showed that ERβ is the principal ER in melanoma, indeed it was found to be the predominant ER type in melanocytic lesions both benign and malignant. Furthermore, they observed that all melanocytic lesions expressed both ER-α and ER-β mRNA and ER-β protein but their expression decreased with the melanoma progression and invasiveness. Moreover, when the tumor lesions were divided into two groups, according to the Breslow thickness index, ER-β mRNA and ER-β protein were found at lower levels in thicker, more invasive tumors. So, the authors concluded that in melanoma the evolution to metastasis could depend on one step where the progression of the tumor becomes independent of the ER system as a result of the loss of the ER-β receptor [27,28]. Another study showed that the incubation in vitro with 17-β-estradiol (E2) stopped human metastatic melanoma cells growth with consequent interleukin-8 (IL-8) mRNA reduction. However, it was observed that the growth inhibition by E2 was countered by exogenously added IL-8. The authors concluded that estrogen works as a suppressor by the inhibition of IL-8 expression and that estrogen mediates an inhibitory action on melanoma via ER and IL8, because this effect was observed exclusively in ER (+) cells and not ER (-) cells [29,30]. In vivo studies on mice highlighted that the use of 2-methoxyestradiol (2ME2), a non-toxic endogenous metabolite of E2, blocks the human melanoma cell-cycle, inducing apoptosis. Sex-related differences in metastasis formation were principally observed in the liver, the main organ active in estrogen conversion into 2ME2. These studies confirmed that the estrogen and its metabolites appear able to exert a direct inhibitory activity on melanoma cells and can have an indirect inhibitory effect via influencing the tumor microenvironment [31,32,33]. However, the role of hormonal receptors in the pathogenesis of melanocytic lesions still remains unclear. A cross–sectional study by Spałkowska et al., aiming to assess ER-α, ER-β and GPER expression on melanocytes and keratinocytes of common nevi, dysplastic nevi, and healthy skin margin in 73 consecutively excised melanocytic lesions, showed, by immunochemistry analysis, the lowest ER-β expression in melanomas and dysplastic nevi, confirming a previous study of Giorgi et al. who reported a significantly lower ER-β expression in melanoma tissue compared with adjacent healthy skin [34]. In a detailed review, Bhari et al. described the fundamental role of estrogens in melanoma evolution. They concluded that, because the effect of estrogen signaling on a tissue is strictly dependent on ER-α and ER-β expression, gender differences in melanoma could be actually linked to the different expression of ERs [35,36,37]. Anyway, it is also clear that many aspects on this topic still remain to be elucidated.

### 2.2. Melanoma and Testosterone

The androgen receptor (AR) belongs to the nuclear receptor superfamily and many reports demonstrated an AR involvement in melanoma growth and invasion in vitro, whereas a clear correlation between its expression and melanoma outcome in vivo is lacking. Unlike estrogens, consistently recognized as a protective factor against melanoma, much less is known about male sex hormone signaling in skin cancer. Nevertheless, epidemiological data support the hypothesis that T might have a clinical role in melanoma. Several studies showed that T, the most abundant androgen in males, promotes different cell types proliferation, including fibroblasts [38], visceral preadipocytes [39], glioblastoma-derived cells [40] and melanoma [41] through the AR. Although AR signaling has been largely studied in tumorigenesis in prostate, breast, bladder, kidney, lung, and liver, little is known about its role in melanoma. For the first time in 1980, Rampen and Mulder proposed that the lower survival of male patients with melanoma could be explained also by the differences in the androgen levels [42]. Subsequently, in 1995, the implication of functional AR in human melanoma cell line IIBMEL-J was proved by growth inhibition in vitro and in vivo with antiandrogens [43]. The authors observed that in vitro incubation with androgens significantly stimulated cell proliferation which could be reversed by the use of hormone receptor antagonists such as flutamide (FLU). These results were confirmed by in vivo experiments on nude mice, transplanted with IIB-MEL-J tumor, where the use of FLU caused an inhibitory effect on cell growth and tumor growth and a remarkable increase in survival rates. The same authors investigated the presence of AR in two other human melanoma cell lines, IIB-MEL-LES and IIB-MEL-IAN, as well as in biopsies from human metastatic melanoma. The presence of AR was confirmed in both cell lines by immunocytochemistry and several hormones and anti-hormones were tested for their ability to affect cell proliferation. In both cell lines, T, DHT, estradiol and progesterone significantly stimulated cell proliferation, and this was reversed by FLU, bicalutamide or tamoxifen (hormone receptor antagonists) [44]. Accordingly, Allil et al. demonstrated in vitro, on cultured murine melanoma cells, the tumor growth and melanogenesis after T dose-dependent incubation, showing that also light exerts a pronounced regulatory effect on tumor growth and a possible interaction with androgens [43]. A study based on analysis of clinical samples and melanoma cells from male and female patients showed that genetic and pharmacological suppression of AR activity triggers melanoma cell senescence and limits tumorigenesis, while increased AR expression or activation exert opposite effects. In fact, AR down-modulation elicits a gene expression signature associated with better patient survival, related to interferon and cytokine signaling and DNA damage/repair. AR down-modulation or pharmacological inhibition suppress melanoma genesis, through an increase in intratumoral infiltration of macrophages and, in an immune-competent mouse model, cytotoxic T cells. [45]. Finally, Watts et al., reported an association between T plasma levels and malignant melanoma in men. This study aimed at examining the associations of serum concentrations of free and total T with the 19 types of cancer in a cohort of 182,600 men and 122,100 postmenopausal women in the UK Biobank, and showed that higher T concentrations were associated with a higher risk of melanoma and prostate cancer in men [46]. Thanks to Aguirre-Portolés C and collaborators, an alternative testosterone pathway has been identified for the first time: studying 98 human melanocytic lesions (nevus, primary, and metastatic melanoma from both males and females) they reported that testosterone promoted melanoma proliferation through activation of ZIP9 (SLC39A9), a zinc transporter that is widely expressed in human melanoma, but not yet targeted by available therapeutics [47] (Figure 1).

## 3. Melanoma and Immunotherapy

For several decades, the treatment of advanced cancer has been challenged by the lack of reliable treatment options and alternatives to chemotherapy, which is often associated with adverse events and high relapse rates [48]. Melanoma, similar to many other tumors, is a chemoresistant tumor, making some previous treatments difficult and often ineffective. However, the recent development of successful alternatives to classic cancer treatments such as immune checkpoint inhibitors (ICIs) and targeted therapies are dramatically changing the treatment landscape also for melanoma patients. New acquisitions in the field of immunology have made it possible to develop therapies able to eliminate cancer by activating the immune response. Immunotherapy led to an improvement of the prognosis of many patients with a wide variety of hematological and solid malignancies. It is based on immunotherapeutic agents capable of activating or boosting the immune system response, in order to reduce off-target effects of chemotherapy and directly kill cancer cells through physiological mechanisms often evaded in the offensive phase of the disease [49]. The effective approach for the activation of antitumor immune responses is based on their modulation through the use of monoclonal antibodies, ICIs, directed towards specific control points (checkpoints) for the activation and development of T cells [50,51,52]. The most commonly observed targets on activated T cells and the most reliable for cancer treatment are cytotoxic T-lymphocyte associated protein 4 (CTLA-4), programmed cell death protein 1 (PD-1) and its PD- ligand. The PD-1/PD-L1 axis plays a key role in the escape of cancer from immunosurveillance; in fact, PD-1 is highly expressed in effector T cells present in the tumor microenvironment and PD-L1 is expressed in cell surfaces in different types of cancer including bladder, lung, colon, breast, kidney, ovary, cervix, melanoma, glioblastoma, multiple myeloma and lymphoma T-cell [53]. The blocking of these checkpoints, thanks to the use of ICIs, has been the most successful strategy to date to stimulate the antitumor immune response. Lymphocyte-activation gene 3 (LAG-3) is a cell surface molecule that is expressed on immune cells, including T cells, and negatively regulates T-cell proliferation and effector T-cell function. LAG-3 is upregulated in many tumor types, including melanoma. Similar to PD-1, LAG 3 is an inhibitory immune checkpoint and it is expressed on tumor-infiltrating lymphocytes. To date, there are six drugs approved by the US FDA, for the immunotherapy treatment of different types of advanced cancer, three PD-1 blockers (pembrolizumab, nivolumab and cemiplimab), its PD-L1 ligand (atezolizumab, avelumab and durvalumab) and a CTLA-4 targeting drug (ipilimumab), but many others are still under study [54,55,56]. ICI therapy drastically transformed the management of advanced melanoma and of melanoma at high risk of recurrence. After the introduction of immunotherapy the average life expectancy for a patient with metastatic melanoma, ranging from six to twelve months before, has definitely improved: in patients treated with anti–PD-1 alone or in combination with ipilimumab the 3-year overall survival (OS) exceed 50% [57], reaching 35–40% at 5 years for anti–PD-1 alone [58] and over 50% at 6.5 years for nivolumab plus ipilimumab [59]. The success of combined therapies has encouraged multiple clinical trials for other cancers, and their effectiveness has been widely proven, nonetheless it is influenced by numerous factors: the immune response, the intrinsic characteristics of cancer cells and the environment, in addition to having frequently adverse effects including non-specific inflammation and autoimmunity. Indeed, ICIs that alter the immune response trough T cells inhibition can lead to a spectrum of inflammatory side effects, as immune-related adverse events (irAEs), caused by pathways involving autoreactive T Cells, autoantibodies and cytokines [60]. In studies evaluating the safety profile of these treatments, it has been reported that more than 60% of patients develop side effects that may involve any organ leading to thyroiditis, hepatitis, pneumonia, hypophysis disease, uveitis, polyneuritis, pancreatitis, colitis, myocarditis and rashes including endocrinopathies (3–23%).

## 4. Sex Disparities and Immunotherapy

Despite the acknowledged sex-related dimorphism in immune system response and in melanoma disease, little is known about the effect of patients’ sex on the efficacy of immune checkpoint inhibitors. Indeed, gender differences are of primary importance in this field, since it seems that they are responsible for a different immunotherapy efficacy between male and female patients. Recent meta-analyses show conflicting data on sex-dependent benefits after systemic treatment for advanced melanoma. Putative causes of these sex disparities are attributable to differences in the immune system, as well as to a role of sex steroid hormones in immunomodulation [61]. Recently, the relevance of sex-dimorphism in the effectiveness of ICIs was highlighted, demonstrating that male and female patients respond in a different way to immunotherapies, regardless of the tumor histological type, the type of treatment and the setting of therapy. Initially, these disparities were ascribed to the well-known differences between male and female immune systems, distinguishing both the immunological response to antigens and innate and adaptive immune responses. However, because some differences are present throughout life, whereas others are typical of puberty or reproductive aging, probably both genetics and hormones are involved [62]. Recently, both biological factors (hormonal and genetic) and sociological (gender difference) have shown a sex-dependent impact on immune function, affecting the antitumor efficacy of the immune checkpoint inhibitors (ICIs) [63,64]. Animal studies showed that sex hormones regulate the expression and function of PD-1 and PD-L1, and that the hormonal effects on the PD-1 pathway are important in mediating autoimmunity [19,65]. In relation to these observations, the different efficacy of an anti-PD-L1 monoclonal antibody has been shown in female compared with male mice in murine melanoma models [20]. Based on existing knowledge, Conforti F et al., demonstrated that immune checkpoint inhibitors can improve overall survival for patients of both sexes in some types of advanced cancers, such as melanoma and non-small-cell lung cancer, but that male patients could derive a larger relative benefit from immune checkpoint inhibitors than female patients [66]. This aspect is mainly related to sex dimorphism in immunity. Indeed, there is evidence that, on average, women mount stronger immune response than men, and this response might reduce their risk of mortality from cancers [61]. However, tumors in women have more efficient mechanisms to evade immune response, thus becoming more resistant to immunotherapy [21]. Furthermore, the susceptibility of women to develop autoimmune disorders could make them more likely to develop immune checkpoint inhibitor-related adverse events [67].

Satisfactory responses have been obtained in the field of immunotherapy have been obtained using autologous T lymphocytes that are engineered to target intracellular antigens through T cell receptors (TCRs) or cell surface antigens through Chimeric Antigen Receptors (CARs). The success of engineered T cell therapy is apparent from clinical trials with CD19-CAR T cells, in patients affected from acute lymphoblastic leukemia and melanoma. In this respect, cancer testis antigens (CTAs) have been considered promising targets for adoptive T cell therapy thanks to their restricted expression in somatic normal tissues, re-expression in many cancer types, and immunogenic nature [10]. The CTA re-expression was observed in worsening of the disease and was linked in various cancers. Among lCTA MAGE-A3, NY-ESO-1, and PRAME have shown great potential as prognostic biomarkers and immunotherapeutic targets [68,69]. Recently, however, attention has been focused on PRAME, (Preferentially expressed Antigen in Melanoma). PRAME is better classified as a testis-selective and its role in normal and neoplastic cells remain non-understood because may differ depending on its tissue-specificity and/or subcellular localization. In addition to supporting tumor cell features, PRAME has been implicated in the regulation of the immune response. Thanks to this feature, it was identified as a tumor antigen that could be recognized by human leukocyte antigen (HLA)-A*24 cytotoxic T lymphocytes in metastatic cutaneous melanoma [70]. Thanks to its restricted re-expression, PRAME represented a candidate target for cancer treatment and has emerged as a potential candidate target for immunotherapy.

## 5. Endocrinopathies and Immunotherapies

Endocrinopathies are among the most common irAEs and include alterations in thyroid, pituitary, adrenal, and gonadal function and diabetes [71,72,73,74].

A meta-analysis showed a high incidence of all-grade endocrine adverse events related to ICIs therapy for melanoma, non-small cell lung carcinoma, and renal cell carcinoma, further enhanced by combined treatment. The highest incidence of hypophysitis on monotherapy is noted with anti-CTLA-4 therapy with ipilimumab rather than with tremelimumab, although hypophysitis can also develop during PD-1 block. The incidence of hypothyroidism in case of monotherapy is highest for PD-1 inhibitors, followed by PD-L1 and CTLA-4 block, and combined ICIs treatment is responsible of a remarkably higher incidence of hypothyroidism, hyperthyroidism, hypophysitis, and primary adrenal insufficiency when compared to chemotherapy alone [75]. Furthermore, a higher incidence of hypothyroidism was observed in patients with NSCLC, unlike the control group treated with docetaxel [71,72,73,76]. In the light of these data, the evaluation of the history of autoimmune diseases and the awareness of an early diagnosis of irAEs are crucial clinical elements. Although the incidence of irAEs is supported by a significant amount of data, the potential impact of ICIs on gonadal function has not been sufficiently studied. The limited evidence available suggests that ICI-related primary hypogonadism due to orchitis as well as secondary hypogonadism due to hypophysitis are a major risk for infertility. Information coming between 2011 and 2019 from a recent analysis by VigiBase (World Health Organization’s global database of individual case safety reports) highlighted a significant, disproportionately increased risk of hypogonadism in men treated with ICIs. Of the 13 reported cases of hypogonadism, five were classified as secondary and one as primary hypogonadism, and these data were subsequently confirmed by a similar analysis of the French Pharmacovigilance database where among the 249 cases considered with endocrinological disorder associated with ipilimumab, nivolumab or pembrolizumab, 94 (49 females and 45 males) were hypophysitis ICI-related where 8% showed panhypopituitarism with secondary hypogonadism [77,78,79]. A retrospective single-center analysis of patients with melanoma reported that nine of 256 patients had low total T levels, in the absence of hypophysitis. In the expanded access trial, two of 20 patients had transient low T levels, which persisted in only one patient. In the combination ipilimumab + nivolumab trial, low total T levels were observed in four of 45 patients. Most had concurrent non-endocrine irAEs and were receiving high-dose steroids, suggesting that the etiology of the low T during immunotherapies is probably complex and/or multifactorial [80]. A recent study demonstrated a reduction in T levels in 34 out of 49 patients at some point during their treatment. Despite low T levels in two-thirds of patients and a high prevalence (43/49) of symptoms of hypogonadism (i.e., fatigue), only three patients have been treated with testosterone replacement therapy [81]. Importantly, the alteration in sex hormone levels at pituitary or gonadal level could interfere with the efficacy and toxicity of the therapy [82]. Although many aspects need to be clarified to understand the mechanism of possible testis dysfunction during ICIs (direct effect on the testis, secondary effect mediated by pituitary alteration, modifications in sex-hormone binding globulin that alter free T levels, alterations in T metabolism, effects of comorbidities), these data highlight the importance of investigating patients for male hypogonadism and the possible benefit of testosterone replacement treatment [83,84]. In conclusion, patients with stage 3 or 4 melanoma undergoing immunotherapy have an increased risk of developing T deficiency during treatment, leading to the need of constant monitoring of their endocrine function during the first three months of immunotherapy and a follow-up to 12 months after its completion. ICI-related toxicities, besides immune-related adverse events, include [68] endocrine toxicities, immune related hypothyroidism or hyperthyroidism, hypophysitis. These adverse effects have the potential to affect women and men alike. Furthermore, while the gonadotoxic effect of chemotherapy and radiation has been widely demonstrated, still little is known about anti–PD-1, anti-CTLA-4, anti-LAG3, or other ICI therapies effect on hypogonadal axis [70]. The short and long-term effects of these treatments on the female reproductive system are not well understood. However, based on the immunotherapy treatment that is administered, adverse effects on hypothalamic–pituitary–ovarian axis, ovarian function, and conception have been reported. Checkpoint inhibitors, for example, can induce hypophysitis and hypothyroidism, the inhibition of PD-L1 may disrupt normal menstrual cycles and inhibit formation of corpora lutea, kinase inhibitors may disrupt oogenesis, follicular maturation, ovulation, and granulosa response to LH. Imatinib, in particular, may reduce blastocyst development, embryo implantation rates and response to ovarian stimulation, however data to provide adequate characterization of these risks and potential benefits are extremely lacking [85].

## 6. Immunotherapy and Male Fertility

In the context of endocrinopathies related to ICIs, the impact of immunotherapy on the reproductive system, and namely on male fertility, still remains unclear. The interference of ICIs with pituitary gland function can affect the proper functioning of the testes (secondary hypogonadism), with alteration in the endocrine component (testosterone production) and spermatogenesis and therefore infertility [86]. However, other mechanisms of testicular dysfunction might be possible, but in general this area is poorly covered in published studies. A first retrospective work was recently published that evaluates the association of ICIs therapy with testicular function in patients who became sterile after treatment with ipilimumab and nivolumab for more than a month and subsequently died of metastatic melanoma. The analysis of post-autopsy testicular biopsies in six out of seven samples (86%) revealed evident alterations in spermatogenesis including focal active spermatogenesis (*n* = 1), hypospermatogenesis (*n* = 2), and Sertoli cell–only syndrome (*n* = 3). However, due to the limited number of subjects studied, it was premature to conclude that this clinical picture is attributable to ICIs therapy [87]. A monocenter cross-sectional study was performed in Germany on 25 men with an age range of 26–59 years undergoing immunotherapy for melanoma or cutaneous malignant tumors, in order to assess the prevalence of male infertility after ICI treatment [88]. At a single timepoint (median of 20 months after treatment initiation) fertility was investigated by semen analysis, plasma concentration of sex hormones and questionnaires on sexual function. All patients reported normal sexual function and the majority of patients remained fertile during and after the treatment, while four of 25 patients had altered semen parameters; three of them, however, had significant confounding factors (history of testis radiation, alcohol abuse, chemotherapy, bacterial orchitis). Therefore, only one case of azoospermia out of 25 patients seemed to be ICI-related. The authors suspected an autoimmune orchitis, as the cause of spermatogenesis impairment. A significant worsening of seminal parameters has been moreover reported in another patient enrolled in this study, although a reduction in seminal parameters was present since before the treatment. A case report testifies how treatment with Ipilimumab/Nivolumab can negatively interfere with male fertility. The case concerns a normozoospermic 30-year-old patient treated for BRAF negative stage IV metastatic melanoma, who became azoospermic after the treatment and testicular biopsy demonstrated a Sertoli cell-only syndrome [89]. The specific antigen that may be targeted during an autoinflammatory attack in the testis remains currently unknown. Cancer-testis antigens are shared between the tumor and the testes, especially in melanoma, which may lead to autoimmunity [90]. The testis locally generates an efficient innate immune system against pathogens both through immunological cells such as macrophages, T lymphocytes and dendritic cells (DCs) located between the interstitial spaces, and through testicular somatic cells with specific immunological function such as Sertoli, Leydig and peritubular myeloid cells (MPCs) [91]. In fact, testicular cells express and release numerous immunosuppressive molecules such as androgens, PDL-1, Fas ligand (FasL), growth arrest-specific gene product 6 (Gas6), and protein S (ProS), which play an important role in regulating the immune response in loco [92]. The Leydig cells, through the production of androgens, which might act as immunosuppressive molecules, can directly regulate the expansion of testicular macrophages and the number of lymphocytes. It is evident that the possible negative influence of ICIs on testis function, including male infertility, needs to be addressed in future studies to understand the actual risk of testis impairment and the mechanisms involved (primary testicular dysfunction, secondary hypogonadism, interference with hormonal balance due to modifications of SHBG concentration). Semen cryopreservation is a well-established technique for the preservation of fertility in male cancer patients before any antineoplastic treatment. Nevertheless, it is not yet routinely discussed with patients undergoing immune-based treatments. We indeed suggest that male patients should be informed about cryopreservation before ICI treatment and regular andrologic follow up, as long-term organ dysfunctions during these treatments were observed and permanent loss of fertility may be possible [93] (Figure 2).

## 7. Conclusions

The melanoma survival disadvantage male has been recently and extensively showed by numerous studies performed on data collected from SEER and US SEER database in large number of cohorts [7] and supported by large clinical trials. Indeed, many epidemiological data confirm that females are more advantaged in both disease progression into metastases and mortality rates. Despite major progress in melanoma treatment with the development of immunotherapy, some points remain unclear and the male sex is lagging in terms of treatment and understanding. Uncertainty exists on the effect of ICIs on testicular function, in terms of both testosterone production and male fertility. Preliminary studies have shown effects of this treatment in inducing male hypogonadism, both as a consequence of pituitary dysfunction and as a target for autoimmune attack. Furthermore, preliminary data have been reported about the possible role of these treatments in causing spermatogenesis impairment. As a consequence, it is crucial to perform a proper oncofertility counseling at the time of melanoma diagnosis in all patients of reproductive age, in order to inform on both the risk of treatment-related gonadotoxicity and on the possibility to preserve the fertility through sperm cryopreservation before starting anticancer immunotherapies.

## Figures and Tables

**Figure 1 ijms-24-00599-f001:**
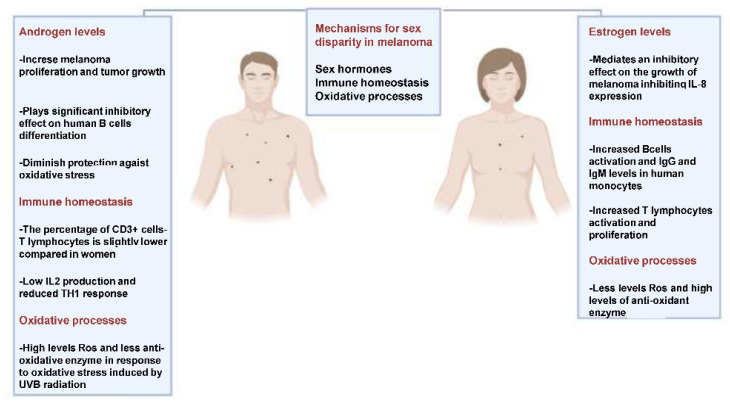
The main underlying mechanisms sex disparities in melanoma [8,14,15,16,23,24,25,26,27,35].

**Figure 2 ijms-24-00599-f002:**
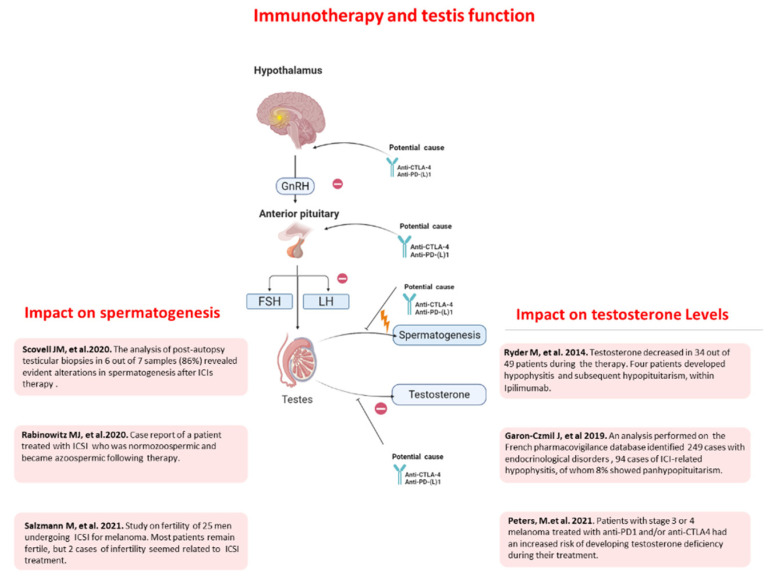
The possible effects direct and indirect of immunotherapy treatment on spermatogenesis and testosterone levels [79,80,82,86,87,89].

## Data Availability

Not applicable.

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
