# Peer review of "Cutaneous Melanoma and Hormones: Focus on Sex Differences and the Testis"

_ijms, 2022, doi:10.3390/ijms24010599_

Round 1
Reviewer 1 Report
1. The role of sex on immunotherapy response is not adequately described. Recent evidence suggest that females respond less to standarf-of-care treatment for advanced melanoma.
2. Any evidence about the role of sex on ICI-induced toxicity?
3. Any evidence about the influence of ICIs on female fertility?
4. Please reduce sections 3+4, as not relevant for the current topic of this review.
5. How does the pre- and postmenopause age influence the sex disparities in melanoma incidence and survival?
6. Please address future research areas
Author Response
- The role of sex on immunotherapy response is not adequately described. Recent evidence suggest that females respond less to standarf-of-care treatment for advanced melanoma.
Answer
Despite the acknowledged sex-related dimorphism in immune system response and in melanoma disease, little is known about the effect of patients’ sex on the efficacy of immune checkpoint inhibitors. Indeed, gender differences are of primary importance in this field, since it seems that they are responsible for a different immunotherapy efficacy between male and female patients. Recent meta-analyses show conflicting data on sex-dependent benefits after systemic treatment for advanced melanoma. Putative causes of these sex disparities are attributable to differences in the immune system, as well as to a role of sex steroid hormones in immunomodulation. Recently, the relevance of sex-dimorphism in the effectiveness of ICIs was highlighted, demonstrating that male and female patients respond in a different way to immunotherapies, regardless of the tumor histological type, the type of treatment and the setting of therapy. Initially, these disparities were ascribed to the well-known differences between male and female immune systems, distinguishing both the immunological response to antigens and innate and adaptive immune responses. However, because some differences are present throughout life, whereas others are typical of puberty or reproductive aging, probably both genetics and hormones are involved. Recently, both biological factors (hormonal and genetic) and sociological (gender difference) have shown a sex-dependent impact on immune function, affecting the antitumor efficacy of the immune checkpoint inhibitors (ICIs).Animal studies showed that sex hormones regulate the expression and function of PD-1 and PD-L1, and that the hormonal effects on the PD-1 pathway are important in mediating autoimmunity. In relation to these observations, the different efficacy of an anti-PD-L1 monoclonal antibody has been shown in female compared with male mice in murine melanoma models.21 Based on existing knowledge, Conforti F et al., (biblio) demonstrated that immune checkpoint inhibitors can improve overall survival for patients of both sexes in some types of advanced cancers, such as melanoma and non-small-cell lung cancer, but that male patients could derive a larger relative benefit from immune checkpoint inhibitors than female patients. This aspect is mainly related to sex dimorphism in immunity. Indeed, there is evidence that, on average, women mount stronger immune response than men, and this response might reduce their risk of mortality from cancers . However, tumors in women have more efficient mechanisms to evade immune response, thus becoming more resistant to immunotherapy. Furthermore, the susceptibility of women to develope autoimmune disorders could make them more likely to develop immune checkpoint inhibitor-related adverse events .
Any evidence about the role of sex on ICI-induced toxicity?
Answer
ICI-related toxicities, besides immune-related adverse events, include 13 endocrine toxicities, immune related hypothyroidism or hyperthyroidism, hypophysitis. These adverse effects have the potential to affect women and men alike. Furthermore, while the gonadotoxic effect of chemotherapy and radiation has been widely demonstrated, still little is known about anti–PD-1, anti-CTLA-4, anti-LAG3, or other ICI therapies effect on hypogonadal axis.
- Any evidence about the influence of ICIs on female fertility?
Answer
The short and long-term effects of these treatments on the female reproductive system are not well understood. However, based on the immunotherapy treatment that is administered, adverse effects on hypothalamic–pituitary–ovarian axis, ovarian fuction, and conception have been reported. Checkpoint inhibitors, for example, can induce hypophysitis and hypothyroidism, the Inhibition of PD-L1 may disrupt normal menstrual cycles and inhibit formation of corpora lutea, kinase inhibitors may disrupt oogenesis, follicular maturation, ovulation, and granulosa response to LH. Imatinib, in particular, may reduce blastocyst development, embryo implantation rates and response to ovarian stimulation, however data to provide adequate characterization of these risks and potential benefits are extremely lacking.
4. Please reduce sections 3+4, as not relevant for the current topic of this review
In the text
- How does the pre- and postmenopause age influence the sex disparities in melanoma incidence and survival?
The relation between menopausal status and melanoma survival is conflicted. Reproductive status in women is characterized by fluctuations of sex steroid hormones. During the reproductive-age, women have a more reactive inflammatory profile and higher levels of T lymphocytes when compared to post-menopausal women. However there appears to be no association between pregnancy and melanoma and the correlation between pre and post menopause and survival is contradictory. Indeed some studies and, recently, Enninga EAL, et al, have found no evidence of differences in post-menopausal groups while others a significant differences in post-menopausal groups documented.

Reviewer 2 Report
Thank your for the opportunity to review this narrative review by Cosci et al. on melanoma and hormones.
Overall, I very much enjoyed the read and I believe the manuscript will be of interest. I also support the conclusions based on your review.
Some comments for improvement:
- While general aspects of ICI therapy take up a big amount of the manuscript, I am missing two sections: one on cancer testis antigens, which would be very interesting in this context, and another one on female fertility.
- When reporting past trials, please check for current updates to report (e.g. 4-year-results of CheckMate 067 are described, when there is a 6.5-year-update: doi: 10.1200/JCO.21.02229)
- Please double check typos and minor errors, example: “detrital” cells instead of dendritic cells in line 269
- I would also suggest a moderate English revision.
Author Response
1 While general aspects of ICI therapy take up a big amount of the manuscript, I am missing two sections: one on cancer testis antigens, which would be very interesting in this context, and another one on female fertility.
Answer
Satisfactory responses have been obtained in the field of immunotherapy have been obtained using autologous T lymphocytes that are engineered to target intracellular antigens through T cell receptors (TCRs) or cell surface antigens through Chimeric Antigen Receptors (CARs). The success of engineered T cell therapy is apparent from clinical trials with CD19-CAR T cells, in patients affected from acute lymphoblastic leukemia and melanoma In this respect, cancer testis antigens (CTAs) have been considered promising targets for adoptive T cell therapy thanks to their restricted expression in somatic normal tissues, re-expression in many cancer types, and immunogenic nature [10].The CTA re-espression was observed in worsening of the disease and linked in various cancers .Among al lCTA MAGE-A3, NY-ESO-1, and PRAME have shown great potential as prognostic biomarkers and immunotherapeutic targets However recently attention has been focused on PRAME,(Preferentially expressed Antigen in Melanoma). PRAME is better classified as a testis-selective and its role in normal and neoplastic cells remain non understood because may differ depending on its tissue-specificity and/or subcellular localization. In addition to supporting tumor cell features, PRAME has been implicated in the regulation of the immune response. Thanks to this feature it was identified as a tumor antigen that could be recognized by human leukocyte antigen (HLA)-A*24 cytotoxic T lymphocytes in metastatic cutaneous melanoma Thanks to its restricted re-expression, PRAME represented a candidate target for cancer treatment and was emerged as a potential candidate target for immunotherapy.
2 When reporting past trials, please check for current updates to report (e.g. 4-year-results of CheckMate 067 are described, when there is a 6.5-year-update: doi: 10.1200/JCO.21.02229)
Was changed in the test
